# How Activin A Became a Therapeutic Target in Fibrodysplasia Ossificans Progressiva

**DOI:** 10.3390/biom14010101

**Published:** 2024-01-12

**Authors:** Dushyanth Srinivasan, Martin Arostegui, Erich J. Goebel, Kaitlin N. Hart, Senem Aykul, John B. Lees-Shepard, Vincent Idone, Sarah J. Hatsell, Aris N. Economides

**Affiliations:** Regeneron Pharmaceuticals, Inc., Tarrytown, NY 10591, USA; dushyanth.srinivasan@regeneron.com (D.S.); martin.arostegui@regeneron.com (M.A.); erich.goebel@regeneron.com (E.J.G.); kaitlin.hart@regeneron.com (K.N.H.); senem.aykul@regeneron.com (S.A.); john.leesshepard@regeneron.com (J.B.L.-S.); vincent.idone@regeneron.com (V.I.); sarah.hatsell@regeneron.com (S.J.H.)

**Keywords:** heterotopic ossification, activin, ACVR1, fibroadipogenic progenitor, BMP

## Abstract

Fibrodysplasia ossificans progressiva (FOP) is a rare genetic disorder characterized by episodic yet cumulative heterotopic ossification (HO) of skeletal muscles, tendons, ligaments, and fascia. FOP arises from missense mutations in Activin Receptor type I (ACVR1), a type I bone morphogenetic protein (BMP) receptor. Although initial findings implicated constitutive activity of FOP-variant ACVR1 (ACVR1^FOP^) and/or hyperactivation by BMPs, it was later shown that HO in FOP requires activation of ACVR1^FOP^ by Activin A. Inhibition of Activin A completely prevents HO in FOP mice, indicating that Activin A is an obligate driver of HO in FOP, and excluding a key role for BMPs in this process. This discovery led to the clinical development of garetosmab, an investigational antibody that blocks Activin A. In a phase 2 trial, garetosmab inhibited new heterotopic bone lesion formation in FOP patients. In contrast, antibodies to ACVR1 activate ACVR1^FOP^ and promote HO in FOP mice. Beyond their potential clinical relevance, these findings have enhanced our understanding of FOP’s pathophysiology, leading to the identification of fibroadipogenic progenitors as the cells that form HO, and the discovery of non-signaling complexes between Activin A and wild type ACVR1 and their role in tempering HO, and are also starting to inform biological processes beyond FOP.

## 1. Introduction

Fibrodysplasia ossificans progressiva (FOP, MIM 135100) is a rare autosomal dominant disorder characterized by episodic and progressive heterotopic ossification (HO) of select skeletal muscles, tendons, ligaments, and fascia [1,2,3]. The heterotopic bone forms via an endochondral process and is cumulative as it cannot be surgically removed without recurrence [4,5]. Recent natural history studies provide extensive information on the clinical progression of HO in FOP [3,6,7]. HO in FOP has a pharmacological correlate in that the delivery of certain Bone Morphogenetic Proteins (BMPs) adsorbed into a matrix induces the formation of heterotopic bone via an endochondral process when injected into skeletal muscle [8,9]. This correlation fostered the notion that a specific BMP might be the ‘culprit’ factor that drives HO in FOP. Indeed, early explorations into the molecular mechanisms of FOP implicated BMP4 as that factor [10,11,12]. Histological comparisons of heterotopic bone in FOP with that of BMP-induced heterotopic bone found close similarity between the two during their different stages of formation [13,14], which was considered consistent with the idea that BMPs are driving the process of HO in FOP. Although the general presumption that FOP is a disorder of BMP signaling has stood the test of time, the identification of the precise mechanism of the inappropriate activation of BMP signaling in FOP remained elusive until the present era.

A major step towards understanding FOP was made in 2006. Activin Receptor type 1 (*ACVR1*, MIM 102576), a type I BMP receptor, was identified as the causative gene [15], further supporting the notion that improper activation of BMP signaling underlies HO in FOP. More importantly, it provided ACVR1 as a potential therapeutic target and simultaneously presented a concrete starting point for investigating the molecular mechanisms that drive the pathophysiology of FOP. Like all other type I BMP/TGFß receptors, ACVR1 is a single-pass transmembrane protein; its extracellular domain interacts with ligands, whereas its intracellular domain is comprised of a Serine–Glycine-rich region (commonly referred to as the GS domain) and a Serine/Threonine kinase region. Studies into the function of this gene have already revealed its importance during development, consistent with its extremely high level of evolutionary conservation [16]. 

The first and most common FOP-causing variant to be identified was ACVR1^R206H^, arising from a missense mutation, c.617G>A, p.R206H. Soon thereafter, additional, rarer variants were discovered in small subsets of FOP patients [2,17]. Like ACVR1^R206H^, all these other variants result from amino acid-altering mutations located in the region encoding the cytoplasmic domain of ACVR1. One of these rarer variants, ACVR1^Q207E^ [17], implied a connection to an engineered constitutively active ACVR1, ACVR1^Q207D^ [18,19], strengthening the notion that overactivation of BMP signaling is responsible for HO in FOP. Initial investigations of the molecular properties of ACVR1^R206H^ concluded that it displays a moderate level of constitutive (i.e., ligand-independent) activity [20,21,22] and that it is hyperresponsive to a subset of BMPs, namely BMP2, BMP5, BMP6, and BMP7 [21,22], leading to the proposition that it is these properties that are responsible for the HO-inducing ability of FOP-variant ACVR1 (ACVR1^FOP^). 

However, several lines of evidence appeared inconsistent with this premise. First, if BMPs, along with a certain level of constitutive activity by ACVR1^FOP^, were driving HO in FOP, it would be expected that HO would simultaneously affect multiple anatomic locations and be temporally continuous, whereas it had been reported to be episodic and focal in nature, with only a subset of anatomic locations undergoing HO at any given point in time [1,23]. This discrepancy was further compounded by reports of variable onset of disease activity with subsets of patients developing HO only later in life, as well as periods of quiescence in overall disease activity [6,23,24]. Second, ACVR1 is a key regulator of iron homeostasis [25,26], with genetic ablation of ACVR1 increasing serum iron [25]. Conversely, the overactivation of this pathway results in anemia, such as occurs with loss of function mutations in the negative regulator, TMPRSS6 [27]. Hence, it would be expected that FOP patients should present with chronic anemia if the proposed hyper-responsiveness and/or constitutive activity were physiologically relevant. However, there have been no reports of chronic anemia in FOP patients, except in one patient that carries a known iron refractory iron deficiency anemia (IRIDA)-causing variant of TMRSS6 [28]. Based on these observations, the pathology of HO in FOP appeared more consistent with the activation of ACVR1^FOP^ in response to a ligand that is expressed at anatomical sites undergoing tissue repair (and that are also ‘HO-competent’). 

By eschewing in vitro studies and focusing on disease pathology in FOP mice, we discovered that the culprit ligand responsible for HO in FOP is Activin A [29]. Our findings were independently verified soon thereafter [30]. This discovery resulted in a fundamental revision of our understanding of FOP’s pathophysiology by identifying Activin as a required ligand for HO in FOP and excluding a key role for BMPs (or any level of constitutive activation). Most importantly, these findings provided the impetus to develop a novel disease-modifying therapy for blocking HO in FOP, resulting in clinical trials to test an anti-Activin A monoclonal antibody (REGN2477, garetosmab) as a potential therapeutic [31,32]. Moreover, our investigations into how Activin A specifically activated FOP-variant ACVR1 (but not wild type ACVR1) have necessitated a reevaluation of ACVR1’s and Activin A’s roles in BMP/TGFß signaling outside of FOP.

Our review focuses on the key discovery of Activin A as an obligate ligand for HO in FOP. First, we describe the thinking that led to this discovery and discuss how it has resulted in a radical revision of the molecular mechanism that underlies HO in FOP by effectively dispelling the notion that HO is driven either by the constitutive activity of ACVR1^FOP^ or by its hyperactivation by BMPs. Furthermore, we summarize the considerations that went into the clinical development of an anti-Activin A monoclonal antibody—REGN2477, garetosmab—as a disease-modifying drug for FOP and discuss the current state of the program. Then, we describe very recent findings regarding the potential of anti-ACVR1 antibodies as a therapy for HO in FOP and provide perspective into this somewhat controversial topic while also discussing the unanticipated learnings derived from the corresponding studies. Lastly, we place our findings regarding the interaction of Activin A with ACVR1 into a broader perspective for BMP/TGFß signaling in general. 

## 2. Review of the Literature

### 2.1. HO in FOP Requires Activation of ACVR1^FOP^ by Ligand

The identification of ACVR1 as the causative gene in FOP immediately presented a druggable target: As a single-pass transmembrane receptor and Serine/Threonine kinase, ACVR1 can be blocked either by stopping its interaction with activating ligand(s), as can be achieved by antibodies or ligand traps, or by inhibiting its kinase activity, as can be achieved using small molecule drugs. Both these and other strategies have been considered since [33,34]. We chose to explore ACVR1-blocking antibodies as a potential therapy. However, at the time that we began our exploration, the prevailing belief in the field of FOP research was that ACVR1^R206H^ drives HO by either being hyperresponsive to certain osteogenic BMPs or by being moderately constitutively active [20,21,22]. This uncertainty presented a problem: if ACVR1^R206H^ turned out to be constitutively active, then this would limit pharmacological therapeutic options to either drugs that would induce its degradation (or inhibit its expression) or drugs that would inhibit its kinase activity. Conversely, if the process of HO were dependent on the activation of ACVR1^R206H^ by ligand(s), then either the direct inhibition of the culprit ligand(s) or blocking their interaction with ACVR1 could be potential therapeutic strategies, rendering an antibody-based strategy more likely to succeed. Hence, we sought to address whether HO in FOP requires the activation of ACVR1^FOP^ by ligand(s) or whether it is driven by the constitutive activity of ACVR1^FOP^.

Due to the complexity of FOP’s pathophysiology along with uncertainty surrounding the identity of the cells that generate HO in FOP, we surmised that the question of constitutive activation versus ligand dependence could not be answered using in vitro models. Rather, it would require investigation in a genetically and physiologically accurate in vivo model of FOP. A mouse line where the ACVR1^R206H^ variant had been knocked-in (MGI symbol: Acvr1^tm1Emsh^) was promising, as the chimeric mice (resulting from microinjection of *Acvr1^R206H/+^* mouse embryonic stem (ES) cells into recipient blastocysts) displayed skeletal malformations akin to those observed in FOP and developed heterotopic bone via an endochondral process reminiscent of human FOP [35]. However, the utility of these mice was limited as they could only be studied as chimeras because they could not be propagated. These findings indicated that FOP could be modeled in mice but would likely require a conditional strategy. 

Hence, we generated a ‘conditional-on’ mouse model of FOP based on the ACVR1^R206H^ variant and using a FlEx allele design [36] (Figure 1). In this design (MGI symbol: Acvr1^tm2.1Vlcg^), the exon encoding Arginine 206 from mouse *Acvr1* was altered to its Histidine 206 counterpart and, together with flanking intronic sequences, was placed in the antisense strand of *Acvr1*. To retain gene function, a wild-type corresponding sequence from human *ACVR1* was introduced in the sense strand to replace the mutated mouse sequence. These two regions (mouse R206H and human wild type) were flanked with FlEx arrays (i.e., two sets of non-crossreactive Lox sites, LoxP and Lox2372) in a manner such that Cre will delete the introduced human sequence while simultaneously bringing the R206H-encoding mouse exon (and associated intronic regions) to the sense strand, giving rise to an *Acvr1^R206H^* allele. As intended, the resulting mouse line, *Acvr1[R206H]^FlEx/+^*; *GT(ROSA26)Sor^CreERT2/+^*, was phenotypically wild type and could be propagated to generate cohorts. Upon Cre-mediated recombination, the *Acvr1[R206H]^FlEx^* allele is converted to *Acvr1^R206H^*, and the corresponding mice (*Acvr1^R206H/+^*; *GT(ROSA26)Sor^CreERT2/+^*, here on referred to as FOP mice) develop HO in response to injury [29,37], mimicking what is observed in FOP patients. 

Using this model, we first investigated whether ACVR1^R206H^ drives HO in a ligand-independent manner as had been proposed [20,21,22] or whether it requires activation by ligand(s). To do this, we utilized two inhibitors of multiple BMP family members, ACVR2A-Fc and ACVR2B-Fc [38,39], and discovered that if used prophylactically, they could completely block HO. In parallel, using cultured cells, we demonstrated that ACVR1^R206H^ does not display constitutive activity and traced the Smad1/5/8 signal present under culture conditions without exogenously added ligand to ligands endogenously expressed by the cells, in line with findings published later [30]. These results nullified the idea that ACVR1^R206H^ is constitutively active while clearly indicating it must be activated by BMP/TGFß family ligands to bring about HO. 

### 2.2. The Ligand That Drives HO in FOP Is Activin A

These data provided the impetus to generate ACVR1-blocking antibodies but also presented a new question: which of the ligands that are inhibited by either ACVR2A-Fc or ACVR2B-Fc could be driving HO in FOP? Of the multiple ligands inhibited by ACVR2A-Fc and ACVR2B-Fc, Activin A stood out because of its connection to inflammation and tissue repair [40], along with the fact that ACVR1, as the name suggests, had been initially identified as an Activin A type I receptor through binding studies [41,42,43]. This set of clues led us to test whether Activin A can induce HO or, conversely, whether the inhibition of Activin A, using the monoclonal antibody REGN2477 [44], could inhibit HO in FOP mice. The implantation of a matrix with adsorbed Activin A resulted in the ossification of the implant only in FOP mice and not in wild-type littermates, whereas REGN2477 completely inhibited HO when dosed prophylactically [29]. The results of these experiments unequivocally established Activin A as both necessary and sufficient for HO in FOP.

Furthermore, the beneficial effect of REGN2477 extended to already forming yet nascent (approximately three-week-old) heterotopic bone lesions. In FOP mice, HO continues to develop for more than three weeks following injury. However, administration of REGN2477 in FOP mice three weeks following injury inhibited further growth of the already formed heterotopic bone lesions, blocked the formation of new lesions (as would be expected), and even caused resorption of a subset of small lesions. In contrast, in FOP mice treated with an isotype control antibody, the heterotopic bone lesions continued to expand, and new lesions also arose. These results indicated that Activin A is required not only for the initiation of HO in FOP mice, but also for the continued development of nascent lesions into mature lesions that eventually connect to the skeleton [37]. 

### 2.3. FOP Variant ACVR1 Is Activated by Activin Whereas Wild Type ACVR1 Is Not

Although these results established that Activin A is necessary and sufficient to initiate and support HO in FOP, they did not provide any mechanistic insights into the molecular aspects of signaling. Hence, concurrently with these in vivo experiments, we tested the interaction of Activin A (and multiple other BMP family members) with ACVR1^R206H^ in vitro. Activin A (as well as Activin B, AB, and AC) activates ACVR1^R206H^ and results in Smad1/5/8 phosphorylation, mimicking the signal generated when ACVR1 is engaged by BMPs. In contrast, none of these Activins activate wild-type ACVR1—rather, they form non-signaling complexes (NSCs) with ACVR1 along with its type II receptor partners that can inhibit BMP signaling mediated by wild-type ACVR1 [29,45] (Figure 2). Shortly thereafter, our finding that Activin A activates ACVR1^R206H^ was independently verified and extended to an additional nine FOP-causing variants of ACVR1 [30], firmly establishing the role of Activin A as the culprit ligand for HO in FOP. Multiple independent studies have verified this result since its initial publication [32,46,47,48,49]. 

### 2.4. Inhibition of Activin A Blocks HO in Patients with FOP 

The discovery that Activin A is requisite for HO in FOP, and the fact that its inhibition protects FOP mice from developing heterotopic bone lesions provided the impetus to further test garetosmab in a clinical setting. First, a double-blind, placebo-controlled phase 1 study (NCT02870400) conducted in healthy women of nonchildbearing age demonstrated that garetosmab has a generally acceptable safety profile with no dose-limiting toxicities [31]. Total Activin A levels minimally changed between 4- and 12-weeks following intravenous dosing with 10 mg/kg of garetosmab, suggesting levels at this dose were sufficient to saturate the binding target (Activin A). This cleared the stage to design the phase 2 LUMINA-1 study (NCT03188666) evaluating garetosmab in adult patients with FOP [32]. Given the paucity of detailed FOP natural history information available at the time of study planning, particularly the rate of appearance of new HO lesions in adults with FOP, LUMINA-1 was designed as a double-blind placebo-controlled trial to evaluate the safety, tolerability, and effects of intravenous garetosmab on HO. 

A schematic of LUMINA-1’s design is shown in Figure 3. Briefly, 44 adult patients with FOP (aged 18–60) and a history of FOP disease activity within one year of screening were randomized to receive treatment with garetosmab 10 mg/kg intravenously every 4 weeks (*n* = 20) or placebo (*n* = 24) for 28 weeks (Period 1). This was followed by a 28-week open-label treatment period (Period 2) where patients on placebo crossed over to garetosmab treatment, while patients on garetosmab during Period 1 continued with garetosmab treatment. All patients were given the option to continue garetosmab during a subsequent open-label extension (Period 3). HO lesion activity and volume were assessed using 18F-labelled sodium fluoride positron emission tomography (PET) in combination with low-dose X-ray computed tomography (CT). 

The main safety findings of LUMINA-1 demonstrated that patients treated with garetosmab had a higher rate of epistaxis (50% vs. 16.7%), madarosis (30% vs. 0%), and skin/soft-tissue infections, including acne (60% versus 12.5%), as compared to placebo in Period 1. Most events were considered mild to moderate in severity by the study’s investigators. Five patient deaths occurred during the open-label period of the trial (Periods 2 and 3). The deaths were reported by the study’s investigators as being unrelated to garetosmab and appeared consistent with the known causes of death and life expectancy in patients with FOP of similar age and disease severity; however, a causal relation to garetosmab could not be excluded. 

The primary efficacy endpoint sought to assess the time-weighted percent change in total lesion activity of pre-existing and new HO lesions from baseline to Week 28. This endpoint did not reach statistical significance (−24.6%, CI −51.8–2.5, *p* = 0.07). However, further analysis elucidated that garestosmab primarily prevented the formation of new heterotopic bone lesions, with a ~97% relative reduction in new HO lesion activity (post hoc) in Period 1. In total, 29 new HO lesions were identified by PET in the placebo cohort versus 3 in those treated with garetosmab. These findings were further corroborated by patients who had transitioned from placebo in Period 1 to garetosmab in Period 2, with a 95% reduction in new HO lesions as measured using PET (23 lesions in Period 1 vs. 1 lesion in Period 2, *p* = 0.0039). Similar results were seen when new HO lesions were assessed via CT. The results mirrored that obtained in FOP mice and effectively demonstrated that Activin A is required for HO in patients with FOP. 

In addition, garetosmab reduced the frequency of patient-reported and investigator-reported flare-ups by nearly 51% and 76%, respectively, in Period 1. Flare-ups are an important feature of FOP and can be severely debilitating for patients. Garetosmab’s impact on flare-ups suggests the role of Activin A in this part of FOP’s presentation. Based on these results, garetosmab is now being further evaluated in a phase 3 study, OPTIMA (NCT05394116), in adult patients with FOP.

### 2.5. Anti-ACVR1 Antibodies Activate Signaling by FOP-Variant ACVR1

While the discovery that Activin A is required for HO in FOP provided a path to a potential disease-modifying therapy, alternative strategies toward the inhibition of HO in FOP continued to be explored [33,34]. Once it was clear that HO in FOP is dependent on the activation of ACVR1^FOP^ by ligands rather than constitutive activity, monoclonal antibodies blocking ACVR1’s interactions with its ligands was one of the strategies considered. Several different groups, including our own, derived such antibodies [49,50,51], which blocked ligand induced signaling from ACVR1 and ACVR1^FOP^ in vitro. However, it turned out that, for the ACVR1^FOP^ variant, these initial in vitro results using overexpression systems were not reflective of physiology. When injected into FOP mice, these antibodies not only did not block injury-induced HO but rather exacerbated it, and they were sufficient to sustain HO, even when Activin A was inhibited. When we retested the ACVR1 antibodies in cells where ACVR1^FOP^ was expressed from the endogenous locus (which is more reflective of physiological conditions), the antibodies induced Smad1/5/8 phosphorylation in the absence of ligand, although less efficiently than Activin A. As Fabs derived from these antibodies blocked signaling in vitro and HO in vivo, we surmised that ACVR1^FOP^ is uniquely activated by dimerization by dimeric ACVR1 antibodies and not just its natural ligands. This was confirmed in orthogonal experiments, in which we demonstrated other means of dimerizing ACVR1^FOP^ also result in the activation of signaling.

The fact that dimerization of a type I receptor was adequate for signaling appeared contradictory to the well-established understanding that type II receptors (and their kinase activity) are required for signaling [52,53,54]. Hence, we surmised that for dimerization to drive activation, the type II receptors must exist in ligand-independent preformed complexes with ACVR1. Indeed, immunoprecipitation of ACVR1 also brought down ACVR2B and vice versa. Similar results were obtained when we tested other type I receptors, such as BMPR1A (ALK3) and BMPR1B (ALK6). Similarly, antibody-mediated activation of ACVR1^FOP^ was abolished by genetic loss of the type II receptors ACVR2A and ACVR2B. Hence, the association of type I and type II receptors in preformed, most likely heterodimeric, and ligand-independent complexes appears to be a shared property of this class of receptors, as has been suggested previously [55,56,57].

In addition to demonstrating that ACVR1 exists in preformed complexes with type II receptors, we also made a few other interesting observations during these experiments. The first was that injury of skeletal muscle was still required for initiation of HO in FOP mice treated with ACVR1 antibodies, indicating that some type of ‘priming’ is required for the cells that generate the heterotopic bone (see below) before they can respond to ACVR1 signaling. The mechanism that brings about this priming is currently unknown. Still, these results are consistent with what had been observed with a transgenic mouse expressing ACVR1[Q207D], an engineered constitutively active and ligand-independent variant of ACVR1. In these mice, injury is also required to induce HO [21], indicating that ACVR1 signaling alone is not adequate to bring about HO. Rather, the cells that assume an osteogenic lineage to generate the heterotopic bone must be primed to respond. The second was that in the presence of ACVR1 antibodies, heterotopic bone lesions in FOP mice mature more slowly but remain in the active state longer than what is seen with isotype control (placebo) treatment. We attributed this to weaker but temporally more sustained signaling by ACVR1 antibodies compared to endogenous Activin A.

Aside from these additional findings, these results indicated that ACVR1 antibodies, unless used as Fabs, are not suitable therapeutics for FOP. More recently, a rat monoclonal antibody (Rm0443) was described that blocks ACVR1 signaling via an indirect mechanism that does not involve competition for binding with a ligand [49]. Rather, Rm0443 appears to inhibit signaling of human ACVR1^FOP^ by holding two ACVR1 molecules in a back-to-back configuration. Nonetheless, consistent with earlier studies [50,51], Rm0443 stimulated signaling of mouse ACVR1^R206H^ and increased the level of HO in FOP mice [49]. However, the authors claimed that this property was restricted to mouse ACVR1^R206H^, whereas human ACVR1^FOP^ was inhibited by Rm0443. They traced this diametrically opposite response to a single amino acid difference in the cytoplasmic domain of human ACVR1 compared to mouse—amino acid 330—which is Serine in mouse ACVR1, versus Proline in human ACVR1. Using cell lines overexpressing different variants of ACVR1, they demonstrated that mouse ACVR1^R206H^ that has also been rendered 330P (mouse ACVR1^R206H;S330P^) became resistant to activation by Rm0443, mirroring what is observed with human ACVR1^R206H^. Finally, they showed that in a new mouse model of FOP that utilizes human ACVR1^R206H^ rather than mouse, Rm0433 appeared to block HO.

In as much as these data appears promising, we postulate an alternative model that is also consistent with the evidence, where the apparent inhibition of HO in the human ACVR1^R206H^ model (*hALK2(R206H) FlEx KI*; *CAG-cre/Esr1*) by Rm0443 results from the poor activation of signaling by this antibody. Overall, ACVR1 antibodies are much less efficient ligands than Activin A and activate signaling via ACVR1^R206H^ to a much lower level. This effect is further compounded by the greatly reduced kinase activity of human ACVR1 [50]. In addition, the model utilized in this study suffers from two methodological drawbacks: that the time frame of monitoring HO was only three weeks post injury, and that this model is particularly mild, developing much less heterotopic bone than an identically engineered *Acvr1^R206H^* model where the mouse receptors is utilized (*mAlk2(R206H) FlEx KI*, activated using adenovirus-delivered Cre) [49]. Therefore, until testing of Rm0443 is repeated in more severe model of FOP (that is based on human ACVR1), the reported inhibition by this antibody should be viewed with caution.

### 2.6. Fibroadipogenic Progenitors Are the Heterotopic Bone-Forming Cells in FOP

The discovery that Activin A is required for HO in FOP provided the impetus to revisit some long-standing questions about FOP’s pathophysiology. One of the main questions in the FOP field had been the identity of the cells that form the heterotopic bone. Identifying such cells could potentially enable a more detailed investigation into the mechanistic aspects of HO in FOP and perhaps identify additional pathways that could be targeted for therapeutic interventions.

Early experiments established that HO in FOP proceeds via the endochondral pathway, as cartilage was found in actively growing lesions that had been resected from FOP patients [13,14]. (It should be reiterated that surgical resection of heterotopic bone is contraindicated in FOP, as it invariably results in the formation of new heterotopic bone [4,5]). The presence of a cartilage intermediate was confirmed using mouse models of FOP [13,29,35,37]. These experiments also showed the presence of other cell types in pre-osseous lesions but did not identify which cell type(s) gave rise to lesion cartilage.

In order to identify which cells might be generating the heterotopic bone, Dey et al. [46] utilized a genetic approach using tissue-specific Cre lines crossed to a constitutively active ACVR1 transgenic line, Tg(CAG-LacZ,-ACVR1*,-EGFP)35-1Mis (also referred to as *Acvr1^Q207D-Tg/Q207D-Tg^*) [19]. The Cre lines were chosen based on expression in cell types that had been considered potential contributors to HO in FOP. In doing so, endothelium, pericytes, myogenic cells, and numerous other connective tissue-resident cell types were conclusively excluded as direct contributors to HO in FOP, invalidating prior claims to the contrary [58]. Of the eleven Cre drivers screened, only two resulted in HO, Scx-Cre, which presented spontaneous HO in ligaments, tendons, and knee joints, and Mx1-Cre, which developed injury-dependent HO in skeletal muscles. The differences in HO location and cause of onset reflect that Scx identifies tenogenic progenitors found in tendons and ligaments [59], whereas Mx1-cre labels a subset of skeletal muscle interstitial cells [60]. Although spatially distinct, both cell populations share functionalities such as ECM secretion and adipogenic potential [61]. These observations led the authors to speculate that these MX1+ and SCX+ cells could be categorized into an overall class of progenitors known as fibroadipogenic progenitors (FAPs) [62,63,64,65]. These progenitors have been shown to possess osteogenic potential in vivo [66,67] and are required for effective repair of muscles [68] and tendons [69] but do not contribute to the remodeled tissue directly. In that sense, they normally function as a transient cell type that is required for repair, but which is not incorporated into the new tissue.

The identification of FAPs as the cell that gives rise to HO in FOP was independently verified and further elaborated upon by Lees-Shepard et al. [47]. They first developed a new conditional-on mouse model of FOP, *Acvr1^tnR206H/+^* (MGI symbol: Acvr1^tm1Glh^), which allows tracking of the cells where the *Acvr1^tnR206H^* allele has been recombined by Cre to generate *Acvr1^R206H^* and simultaneously express eGFP. In this manner, any cell that carries the FOP-causing genotype can be tracked and isolated. Using a Tie2-Cre driver, which is expressed in a subset of FAPs, they demonstrated that the resulting mice develop HO in response to injury. In an orthogonal series of transplantation experiments, they isolated PDGFRa+ cells (presumptive FAPs) from *Acvr1^tnR206H/+^*; *Tie2-Cre* mice and demonstrated the FAPs generated heterotopic bone when implanted into the muscle of SCID mice. The heterotopic bone formed via the endochondral pathway, mirroring what has been observed in FOP, and was dependent on Activin A. Notably, the inhibition of Activin A restored wild-type behavior to FAPs isolated from FOP mice. These results unequivocally indicated that FAPs are the only cells necessary to replicate the pathology of HO in FOP and excluded potential contributions from other muscle-resident cells such as satellite cells and endothelial cells. Based on the shared expression of cell surface markers and overtly similar lineage capacity, the authors proposed that Tie2, SCX, and MX1 lineage-labeled progenitors may be unified under the “FAP” moniker. However, many questions remain regarding the heterogeneity of anatomically distinct “FAPs”.

### 2.7. Wild-Type ACVR1 Forms Non-Signaling Complexes with Activin A and Cognate Type II Receptors

Another observation that arose from the discovery that Activin A uniquely activates FOP-variant was that Activin A must interact with wild-type ACVR1, as had been originally described [41,42,43]. In fact, ACVR1’s name—Activin A receptor, type I—was based on those initial findings. However, the failure of Activin A to activate ACVR1, along with the discovery that ACVR1 normally propagates a Smad1/5/8 signal and responds to BMP7 and the discovery that Activin A initiates Smad2/3 signaling via ACVR1B (also known as ALK4), resulted in a reclassification of ACVR1 as a BMP receptor [70,71,72]. The fact that Activin A activates FOP-variant ACVR1 made it clear that these initial reports that Activin A utilizes ACVR1 as a receptor [41,42,43] were valid, although that binding could only be detected using crosslinking. In spite of this seemingly transient association, Activin A competitively blocks BMP-initiated signaling via ACVR1 [29,45]. Furthermore, this competition is not solely due to Activin A occupying type II receptors [73,74], as hindering the association of Activin A with ACVR1 reduced its ability to block BMP-initiated signaling by about 60-fold, indicating that the association with ACVR1 significantly contributes to the competition. Based on these results, we surmised that Activin A must form non-signaling complexes (NSCs) with ACVR1 along with its partner type II receptors [29,45] and that these NSCs are required for the effective blocking of BMP signaling mediated by ACVR1 [45]. In the case of FOP, this competitive inhibition of signaling via NSCs extends to Activin itself, as in FOP Activin A can form complexes with either ACVR1^FOP^ or wild-type ACVR1. Since complexes with wild-type ACVR1 would fail to signal, they would effectively reduce the amount of Activin A available to interact with ACVR1^FOP^. Hence, we surmised that these ACVR1•Activin A•type IIR NSCs are likely to play a tempering role in the formation of heterotopic bone in FOP. 

### 2.8. The ACVR1•Activin A•Type II Receptor Non-Signaling Complex Tempers the Severity of HO in FOP Mice

To interrogate directly the role of these NSCs, we engineered an ‘agonist-only’ Activin A mutein, i.e., a mutein that can activate Smad2/3 signaling via ACVR1B but which fails to form NSCs with ACVR1 [45]. We achieved this by replacing five amino acids comprising the finger two tip loop region of Activin A with the corresponding region of Nodal. (The finger two tip loop region is a major determinant of type I receptor preference by the Activins and related ligands [75]). The resulting mutein, Activin A.NodF2TL (F2TL), displayed the desired properties in that it retained the ability to activate ACVR1B yet failed to form NSCs with ACVR1. In addition, it displayed another useful property: it was still capable of activating ACVR1^FOP^, presumably because it still dimerizes the preformed heterodimers of ACVR1 plus type II receptor by interacting with the type II receptors [50]. We took advantage of this property to show that the NSCs that Activin forms with wild-type ACVR1 are physiologically relevant in FOP. Since the engineered F2TL mutein escapes from incorporation into NSCs, we predicted that in FOP mice, it should generate more heterotopic bone than an equivalent amount of wild-type Activin A. Indeed, when Geltrex containing adsorbed F2TL was implanted into FOP mice, it generated approximately six times more heterotopic bone (at the site of implantation) compared to Geltrex containing the equivalent amount of Activin A. These results demonstrated that the NSC has a tempering effect on HO in FOP, while simultaneously showing that these NSCs are operational in vivo. Moreover, these results were consistent with evidence showing that the loss of wild-type ACVR1 exacerbates HO in FOP mice [47], whereas its overexpression protects FOP mice from developing heterotopic bone [76]. 

## 3. Conclusions and Future Directions

The discovery of Activin A as the culprit ligand in FOP was the second major step after the discovery of the causative gene, ACVR1, in elucidating the molecular mechanisms that underly the pathophysiology of this disease. This discovery had immediate translational consequences as it provided an additional druggable target for FOP, leading to the clinical development of the investigational Activin A antibody garetosmab [32]. In parallel, it necessitated a reevaluation of the molecular mechanisms that drive HO and other phenotypes of FOP. To begin with, the discovery that HO in FOP requires the activation of ACVR1^FOP^ by Activin and that the inhibition of Activin A completely blocks HO dispensed the notion that hyperresponsiveness to BMP or constitutive activity of ACVR1^FOP^ are driving HO in FOP. Surprisingly, multiple other reviews fail to clarify that HO in FOP does not result from either constitutive activation or hyperresponsiveness of ACVR1^FOP^ to BMPs [33,34,77,78,79,80,81,82] but is rather driven by the activation of ACVR1^FOP^ by Activin A. We posit that this perspective is incorrect, as has been demonstrated by our work [29,37] as well as that of Hino et al. [30], has been independently replicated [47,83,84,85,86], and further buttressed by the results of a recent clinical trial [32]. 

Furthermore, the studies exploring the role of Activin A as well as the utility of anti-ACVR1 antibodies in FOP revealed that the activation of ACVR1^FOP^ alone is not adequate to drive the FAPs down an osteogenic lineage; rather, some other factor(s) is required to ‘prime’ the FAP to a state where the activation of the Smad1/5/8 pathway is perceived as an osteogenic signal [29,46,47,50,51]. This finding has not been discussed in any detail elsewhere, and it is important in that identification of the factor that primes the FAPs to respond to Activin A may provide yet another potential therapeutic target in FOP. 

Lastly, FOP provided the first physiological evidence that the interaction between Activin A and ACVR1 is not artifactual and that ACVR1 is indeed a receptor for Activin A, as had been originally described [41,42,43]. This finding indicated that wild-type ACVR1 forms non-signaling complexes (NSCs) with Activin A, leading to the discovery that these non-signaling complexes have a tempering effect on the level of HO in FOP [45,76]. These data also indicate that the formation of the heterotetrameric complexes comprised of two type I receptors and two type II receptors along with the naturally dimeric BMP/TGFß family ligands is not the sole requirement for signaling; rather, an additional mechanism, such as conformational changes, must be at play. This possibility has been cursorily discussed in the literature [87] but has not been explored experimentally.

As informative as these discoveries have been for understanding FOP, many new questions emanate from these discoveries. For example, the mechanism by which FAPs are primed to assume an osteogenic fate in response to Smad1/5/8 is unknown. The mechanism by which stoichiometrically identical complexes comprised of ACVR1, type II receptors, and Activin A, and where ACVR1 is either wild type or ACVR1^FOP^, bring about diametrically opposite outcomes remains to be elucidated. The physiological roles of the NSCs formed by the Activins with wild-type ACVR1 remain completely unexplored outside of FOP. These new lines of inquiry are a direct byproduct of the discovery of Activin A’s central role in FOP but are likely to provide answers that will inform BMP/TGFß signaling mechanisms in a broader sense. 

## Figures and Tables

**Figure 1 biomolecules-14-00101-f001:**
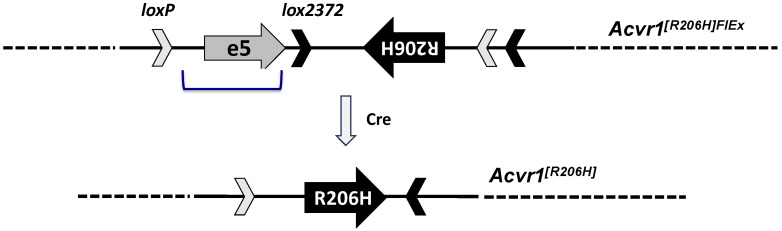
**Simplified structure of *Acvr1[R206H]^FlEx^* allele.** Mouse *Acvr1* exon 5 and flanking intronic sequence were placed in the antisense strand of Acvr1 within intron 5. Mouse *Acvr1* exon 5 was altered to encode R206H. The corresponding human exon and associated intronic sequence (bracketed) were inserted in the sense strand to replace the mouse sequence that has been placed in the antisense strand. These two regions were flanked by FlEx arrays comprised of two lox site, *loxP* and *lox2372* that do not cross-combine. The FlEx arrays were placed in a manner such that, upon action of Cre, the introduced human sequence will be deleted, and the R206H mouse exon region will be brought to the sense strand. In this manner, the corresponding mice start as *Acvr1^[R206H]FlEx/+^*, and hence do not express ACVR1^R206H^, but can be rendered genotypically FOP, i.e., *Acvr1^R206H/+^* by Cre.

**Figure 2 biomolecules-14-00101-f002:**
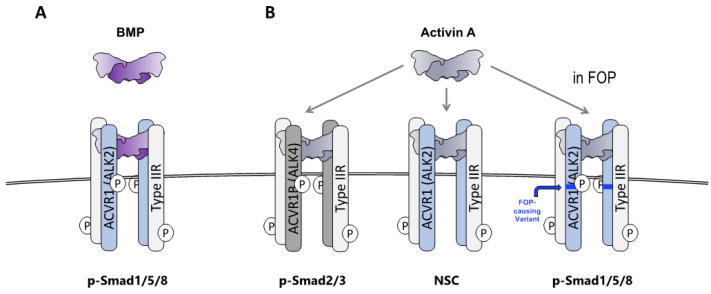
**Activin A activates Smad1/5/8 signaling from ACVR1^FOP^ but generates non-signaling complexes with wild-type ACVR1.** (**A**) When BMPs associate with ACVR1 along with its partner type II receptors (ACVR2A, ACVR2B, and BMPR2), the resulting complex drives the phosphorylation of Smad1/5/8. (**B**) When Activin A associates with ACVR1B and its partner type II receptors (ACVR2A, ACVR2B, and BMPR2), the resulting complex drives phosphorylation of Smad2/3. In contrast, when Activin A associates with ACVR1 and its partner type II receptors, the resulting complex (NSC) does not signal. In FOP, a stoichiometrically identical complex utilizing ACVR1^FOP^ drives phosphorylation of Smad1/5/8, mimicking the action of BMPs.

**Figure 3 biomolecules-14-00101-f003:**
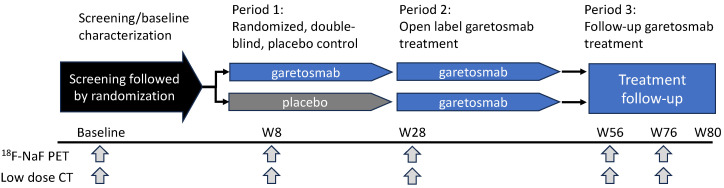
**Overview of LUMINA-1 Study Design**. LUMINA-1 was a phase 2, randomized, double-blind, placebo-controlled study in adult patients with FOP. It consisted of a 4-week screening/baseline period, a 6-month randomized double-blind placebo-controlled treatment period (Period 1), a 6-month open-label garetosmab treatment period (Period 2), and a follow-up treatment period with open-label garetosmab (Period 3). Following confirmed eligibility during the screening/baseline period (Day-28 to Day 1), patients were randomized 1:1 to 10 mg/kg garetosmab IV dosed Q4W or placebo IV. Efficacy was assessed via ^18^F-NaF PET and low-dose CT imaging analysis of HO at weeks 8, 28, 56, and 76.^18^F-NaF = fluorine-18-labelled sodium fluoride; CT = computed tomography; D = day; IV = intravenous; PET = positron emission tomography; W = week.

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
