# Peer review of "How Activin A Became a Therapeutic Target in Fibrodysplasia Ossificans Progressiva"

_biomolecules, 2024, doi:10.3390/biom14010101_

Round 1
Reviewer 1 Report
Comments and Suggestions for Authors
This is a well written and very thorough MS that clearly documents current understanding of FOP. It provides a logical framework for potential paths to disease-modifying therapies. My only suggestions are:
1. Line 180: effects of Activin A on Acvr1 in vitro; I think interaction might be a better choice of words than effects.
2. Line 245: I believe the word through is missing.
3. Line 272: I suggest substituting "while performing these experiments" for during.
4. There are several places, including line 305, where the punctuation does not follow the standard format of colon followed by semicolon followed by comma.
5. Line 340: Alt-hough should be Al-though.
Comments on the Quality of English LanguageThe quality is acceptable.
Author Response
We thank you for providing a positive response to our manuscript and for your encouraging words. We have rendered the request changes. Specifically:
- Line 180: effects of Activin A on Acvr1 in vitro; I think interaction might be a better choice of words than effects.
Revised corresponding sentence to “…the interaction of Activin A (and multiple other BMP family members) with ACVR1R206H…”
- Line 245: I believe the word through is missing.
We could not figure out what word might be missing. The original text states “Monoclonal antibodies to ACVR1 were one of the strategies considered once it was clear that HO in FOP is dependent on activation of ACVR1FOP by ligand rather than constitutive activity.” We have revised to “Once it was clear that HO in FOP is dependent on activation of ACVR1FOP by ligands rather than constitutive activity, monoclonal antibodies blocking ACVR1’s interactions with its ligands was one of the strategies considered.” We hope that this change is acceptable.
- Line 272: I suggest substituting "while performing these experiments" for during.
This correction has been rendered.
- There are several places, including line 305, where the punctuation does not follow the standard format of colon followed by semicolon followed by comma.
It is our interpretation that the reviewer is referring to mouse line genotypes, e.g., “hALK2(R206H) FlEx KI; CAG-cre/Esr1”. The use of a semicolon in reporting such genotypes is standard practice and we (and others) have used extensively in our research manuscripts. We propose to retain this type of nomenclature here in order to simplify cross-referencing to the primary literature.
- Line 340: Alt-hough should be Al-though.
This correction has been rendered.
Reviewer 2 Report
Comments and Suggestions for Authors
This paper lacks original contributions to the FOB field and reiterates information already present in existing reviews. Furthermore, the writing style is unclear and difficult to follow.
Author Response
We respectfully disagree with the Reviewer’s assessment, and particularly with the statement that our manuscript “…lacks original contributions to the FOB field and reiterates information already present in existing reviews”. We wrote our review approaching reviews on FOP from exactly the opposite perspective, that there is no critical review that provides a comprehensive synthesis of the discovery of the role of Activin A in FOP and its implications for understanding FOP, developing therapies, and the greater implications of the reviewed findings for BMP/TGFß signaling in general. We have scoured the literature for such a review, and to the best of our knowledge, no such review exists. Furthermore, we have encountered what we consider to be a serious issue with multiple reviews in that the great majority of them contain erroneous and outdated information on the mechanisms that drive heterotopic ossification (HO) in FOP. More specifically, multiple reviews continue to claim that HO in FOP results from constitutive activation and/or hyperactivation of FOP-variant ACVR1 (ACVR1[FOP]) by BMPs (Cappato et al., 2018; Katagiri et al., 2021; Katagiri et al., 2018; Kitoh, 2020; Lin et al., 2019; Valer et al., 2019; Ventura et al., 2021; Wentworth et al., 2019) and fail to clarify how the discovery of Activin A as a driver of HO in FOP has superseded these prior findings. This perspective is entirely incorrect, as has been proven by our work (Hatsell et al., 2015; Upadhyay et al., 2017) as well as that of Hino et al (Hino et al., 2015), as has been replicated by multiple labs (Lees-Shepard et al., 2018; Mundy et al., 2023; Ramachandran et al., 2021; Wang et al., 2018; Xie et al., 2020). If it were indeed the case that HO in FOP results from constitutive activation and/or hyperactivation of FOP-variant ACVR1 (ACVR1[FOP]) by BMPs, then it follows that inhibition of Activin A would have no effect on HO. As is evident by the work already cited here as well as the results of a recent clinical trial (Di Rocco et al., 2023), inhibition of Activin A is exceedingly efficacious in inhibiting the formation of heterotopic bone in FOP. Hence, first and foremost, our review is an attempt to clarify the point that it is Activin A that drives HO in FOP and that BMPs have no major role to play in that process – no other review that we are aware of states this clearly and unequivocally.
Secondly, as already mentioned, our review provides a synthesis of FOP research as it relates to the role of Activin A, and the new avenues of investigation that this research has opened for FOP researchers. For example, it was the discovery that Activin A activates ACVR1[FOP] that led us to realize that Activin A must form complexes with wild type ACVR1 (as had been originally reported (Attisano et al., 1993; Ebner et al., 1993; Tsuchida et al., 1993) and resulting in the name of this type I receptor – Activin Receptor type 1). This in turn led us and others to the realization that wild type ACVR1-Activin complexes must be incapable of signaling and in fact act as antagonists of signaling in two different levels: (a) antagonize BMP signaling mediated by ACVR1 (Hatsell et al., 2015), and (b) temper the degree of HO in FOP (Aykul et al., 2020; Yamamoto et al., 2022). Again, to the best of our knowledge this aspect of FOP molecular pathophysiology as not been reviewed elsewhere. Moreover, the fact that ACVR1 forms non-signaling complexes with Activin A has been left out of from all recent reviews on TGFß and BMP signaling save one (Martinez-Hackert et al., 2021). Our review rectifies this issue by discussing the role of wild type ACVR1 and its interaction with Activin A in FOP while also pointing out its implications for BMP/TGFß signaling in general.
Thirdly, we review what we and others have learned about the potential of anti-ACVR1 antibodies as a potential therapy for FOP. This is an important subject, since based on our findings (Aykul et al., 2022) as well as those by David Goldhamer’s group (Lees-Shepard et al., 2022), these antibodies exacerbate HO in FOP, and hence their consideration as potential therapeutic should not be considered (Collins, 2022). However, this viewpoint has been challenged by findings from another group (Katagiri et al., 2023). We attempt to provide perspective as to why we maintain our viewpoint that anti-ACVR1 antibodies should not be considered as a therapy. Moreover, we review other learnings that have arisen through the exploration of anti-ACVR1 antibodies, and particularly the discovery that in vivo type I and type II BMP receptors exist as preformed, ligand-independent, non-signaling heterodimers, and we discuss the implications of this finding (Aykul et al., 2022). Again, to the best of our knowledge, none of this work has been reviewed elsewhere.
Lastly, our review is also meant to provide a window into the process of discovery of our findings and how we connect basic biology discovery to drug development. As another the other reviewers noted, our review “…shows how this basic knowledge has been exploited to develop an activin A antagonist for treatment of FOP” and “…provides a logical framework for potential paths to disease-modifying therapies”. Again, to the best of our knowledge no other review has attempted to do this for Activin A in FOP, and we believe that we are uniquely positioned to provide that perspective.
Based on the above, along with the lack of any specific suggestions regarding our writing style, we have not rendered any changes (other than the minor edits suggested by the other reviewers) into our manuscript.
Attisano, L., Cárcamo, J., Ventura, F., Weis, F.M.B., Massagué, J., and Wrana, J.L. (1993). Identification of human activin and TGFβ type I receptors that form heteromeric kinase complexes with type II receptors. Cell 75, 671-680. http://dx.doi.org/10.1016/0092-8674(93)90488-C.
Aykul, S., Corpina, R.A., Goebel, E.J., Cunanan, C.J., Dimitriou, A., Kim, H.J., Zhang, Q., Rafique, A., Leidich, R., Wang, X., et al. (2020). Activin A forms a non-signaling complex with ACVR1 and type II Activin/BMP receptors via its finger 2 tip loop. Elife 9. 10.7554/eLife.54582.
Aykul, S., Huang, L., Wang, L., Das, N.M., Reisman, S., Ray, Y., Zhang, Q., Rothman, N., Nannuru, K.C., Kamat, V., et al. (2022). Anti-ACVR1 antibodies exacerbate heterotopic ossification in fibrodysplasia ossificans progressiva (FOP) by activating FOP-mutant ACVR1. J Clin Invest 132. 10.1172/JCI153792.
Cappato, S., Giacopelli, F., Ravazzolo, R., and Bocciardi, R. (2018). The Horizon of a Therapy for Rare Genetic Diseases: A "Druggable" Future for Fibrodysplasia Ossificans Progressiva. Int J Mol Sci 19. 10.3390/ijms19040989.
Collins, M.T. (2022). Twists in the fibrodysplasia ossificans progressiva story challenge and expand our understanding of BMP biology. J Clin Invest 132. 10.1172/JCI160773.
Di Rocco, M., Forleo-Neto, E., Pignolo, R.J., Keen, R., Orcel, P., Funck-Brentano, T., Roux, C., Kolta, S., Madeo, A., Bubbear, J.S., et al. (2023). Garetosmab in fibrodysplasia ossificans progressiva: a randomized, double-blind, placebo-controlled phase 2 trial. Nat Med. 10.1038/s41591-023-02561-8.
Ebner, R., Chen, R.H., Lawler, S., Zioncheck, T., and Derynck, R. (1993). Determination of type I receptor specificity by the type II receptors for TGF-beta or activin. Science 262, 900-902.
Hatsell, S.J., Idone, V., Wolken, D.M., Huang, L., Kim, H.J., Wang, L., Wen, X., Nannuru, K.C., Jimenez, J., Xie, L., et al. (2015). ACVR1R206H receptor mutation causes fibrodysplasia ossificans progressiva by imparting responsiveness to activin A. Science translational medicine 7, 303ra137. 10.1126/scitranslmed.aac4358.
Hino, K., Ikeya, M., Horigome, K., Matsumoto, Y., Ebise, H., Nishio, M., Sekiguchi, K., Shibata, M., Nagata, S., Matsuda, S., and Toguchida, J. (2015). Neofunction of ACVR1 in fibrodysplasia ossificans progressiva. Proc Natl Acad Sci U S A 112, 15438-15443. 10.1073/pnas.1510540112.
Katagiri, T., Tsukamoto, S., and Kuratani, M. (2021). Accumulated Knowledge of Activin Receptor-Like Kinase 2 (ALK2)/Activin A Receptor, Type 1 (ACVR1) as a Target for Human Disorders. Biomedicines 9. 10.3390/biomedicines9070736.
Katagiri, T., Tsukamoto, S., Kuratani, M., Tsuji, S., Nakamura, K., Ohte, S., Kawaguchi, Y., and Takaishi, K. (2023). A blocking monoclonal antibody reveals dimerization of intracellular domains of ALK2 associated with genetic disorders. Nat Commun 14, 2960. 10.1038/s41467-023-38746-5.
Katagiri, T., Tsukamoto, S., Nakachi, Y., and Kuratani, M. (2018). Recent Topics in Fibrodysplasia Ossificans Progressiva. Endocrinol Metab (Seoul) 33, 331-338. 10.3803/EnM.2018.33.3.331.
Kitoh, H. (2020). Clinical Aspects and Current Therapeutic Approaches for FOP. Biomedicines 8. 10.3390/biomedicines8090325.
Lees-Shepard, J.B., Stoessel, S.J., Chandler, J.T., Bouchard, K., Bento, P., Apuzzo, L.N., Devarakonda, P.M., Hunter, J.W., and Goldhamer, D.J. (2022). An anti-ACVR1 antibody exacerbates heterotopic ossification by fibro-adipogenic progenitors in fibrodysplasia ossificans progressiva mice. J Clin Invest 132. 10.1172/JCI153795.
Lees-Shepard, J.B., Yamamoto, M., Biswas, A.A., Stoessel, S.J., Nicholas, S.E., Cogswell, C.A., Devarakonda, P.M., Schneider, M.J., Jr., Cummins, S.M., Legendre, N.P., et al. (2018). Activin-dependent signaling in fibro/adipogenic progenitors causes fibrodysplasia ossificans progressiva. Nat Commun 9, 471. 10.1038/s41467-018-02872-2.
Lin, H., Shi, F., Gao, J., and Hua, P. (2019). The role of Activin A in fibrodysplasia ossificans progressiva: a prominent mediator. Biosci Rep 39. 10.1042/BSR20190377.
Martinez-Hackert, E., Sundan, A., and Holien, T. (2021). Receptor binding competition: A paradigm for regulating TGF-beta family action. Cytokine Growth Factor Rev 57, 39-54. 10.1016/j.cytogfr.2020.09.003.
Mundy, C., Yao, L., Shaughnessy, K.A., Saunders, C., Shore, E.M., Koyama, E., and Pacifici, M. (2023). Palovarotene Action Against Heterotopic Ossification Includes a Reduction of Local Participating Activin A-Expressing Cell Populations. JBMR Plus 7, e10821. 10.1002/jbm4.10821.
Ramachandran, A., Mehic, M., Wasim, L., Malinova, D., Gori, I., Blaszczyk, B.K., Carvalho, D.M., Shore, E.M., Jones, C., Hyvonen, M., et al. (2021). Pathogenic ACVR1(R206H) activation by Activin A-induced receptor clustering and autophosphorylation. EMBO J, e106317. 10.15252/embj.2020106317.
Tsuchida, K., Mathews, L.S., and Vale, W.W. (1993). Cloning and characterization of a transmembrane serine kinase that acts as an activin type I receptor. Proc Natl Acad Sci U S A 90, 11242-11246.
Upadhyay, J., Xie, L., Huang, L., Das, N., Stewart, R.C., Lyon, M.C., Palmer, K., Rajamani, S., Graul, C., Lobo, M., et al. (2017). The Expansion of Heterotopic Bone in Fibrodysplasia Ossificans Progressiva Is Activin A-Dependent. J Bone Miner Res 32, 2489-2499. 10.1002/jbmr.3235.
Valer, J.A., Sanchez-de-Diego, C., Pimenta-Lopes, C., Rosa, J.L., and Ventura, F. (2019). ACVR1 Function in Health and Disease. Cells 8. 10.3390/cells8111366.
Ventura, F., Williams, E., Ikeya, M., Bullock, A.N., Ten Dijke, P., Goumans, M.J., and Sanchez-Duffhues, G. (2021). Challenges and Opportunities for Drug Repositioning in Fibrodysplasia Ossificans Progressiva. Biomedicines 9. 10.3390/biomedicines9020213.
Wang, H., Shore, E.M., Pignolo, R.J., and Kaplan, F.S. (2018). Activin A amplifies dysregulated BMP signaling and induces chondro-osseous differentiation of primary connective tissue progenitor cells in patients with fibrodysplasia ossificans progressiva (FOP). Bone 109, 218-224. 10.1016/j.bone.2017.11.014.
Wentworth, K.L., Masharani, U., and Hsiao, E.C. (2019). Therapeutic advances for blocking heterotopic ossification in fibrodysplasia ossificans progressiva. Br J Clin Pharmacol 85, 1180-1187. 10.1111/bcp.13823.
Xie, C., Jiang, W., Lacroix, J.J., Luo, Y., and Hao, J. (2020). Insight into Molecular Mechanism for Activin A-Induced Bone Morphogenetic Protein Signaling. Int J Mol Sci 21. 10.3390/ijms21186498.
Yamamoto, M., Stoessel, S.J., Yamamoto, S., and Goldhamer, D.J. (2022). Overexpression of Wild-Type ACVR1 in Fibrodysplasia Ossificans Progressiva Mice Rescues Perinatal Lethality and Inhibits Heterotopic Ossification. J Bone Miner Res 37, 2077-2093. 10.1002/jbmr.4617.
Reviewer 3 Report
Comments and Suggestions for Authors
This review provides a comprehensive discussion of current knowledge related to the etiology of fibrodysplasia ossificans progressiva and shows how this basic knowledge has been exploited to develop an activin A antagonist for treatment of FOP.
Minor points:
Authors should consider changes a few awkward phrases in the text.
ln 287 Aside from these additional findings
ln 302-3. ....we postulate an alternative mosedl that is also consistent with the evidence where
Ln 367. ... regarding the heterogeneity of anatomically distinct "FAPs"...
Comments on the Quality of English Languageminor changes in presentation as noted in comments to authors
Author Response
We thank you for providing a positive response to our manuscript and for your encouraging words. We have rendered the request changes. Specifically:
ln 287 Aside from these additional findings
We have rendered the proposed change into our revised manuscript.
ln 302-3. ....we postulate an alternative mosedl that is also consistent with the evidence where
We have rendered the proposed change into our revised manuscript.
Ln 367. ... regarding the heterogeneity of anatomically distinct "FAPs"...
We have revised our original statement from “However, many questions regarding the true heterogeneity anatomically distinct “FAPs” require further interrogation.” To “However, many questions remain regarding the heterogeneity of anatomically distinct “FAPs”.” We hope that this change makes the reading of the corresponding paragraph easier.
Round 2
Reviewer 2 Report
Comments and Suggestions for Authors
After reading the author's comments, I think the manuscript could be improved by adding a couple of sentences in the introduction that better clarify the novelty of their review. The information that they provide in the response to my comments should be further developed in the discussion section.
Author Response
Thank for these insightful comments. We appreciate your suggestions and have made the following revisions in the text:
- In the Introduction, replaced the concluding statement “In this review we summarize the key findings that led to the clinical development of garetosmab and further connect these learnings to broader aspects of BMP/TGFß signaling.”, with a more inclusive paragraph outlining what our review in meant to accomplish. This paragraph reads as follows: “Our review focuses on the key discovery of Activin A as an obligate ligand for HO in FOP. First, we describe the thinking that led to this discovery and discuss how it has resulted in a radical revision of molecular mechanism that underlies HO in FOP by effectively dispelling the notion that HO is driven either by constitutive activity of ACVR1FOP or by its hyperactivation by BMPs. Furthermore, we summarize the considerations that went into the clinical development an anti-Activin A monoclonal antibody – REGN2477, garetosmab – as a disease modifying drug for FOP and discuss the current state of the program. Then we describe very recent findings regarding the potential of anti-ACVR1 antibodies as a therapy for HO in FOP and provide perspective into this somewhat controversial topic, while also discuss the unanticipated learnings derived from the corresponding studies. Lastly, we place our findings regarding the interaction of Activin A with ACVR1 into a broader perspective for BMP/TGFß signaling in general.”
- We have revised the last section thoroughly to read (see italicized text for new additions):
The discovery of Activin A as the culprit ligand in FOP was the second major step after the discovery of the causative gene, ACVR1, in elucidating the molecular mechanisms that underly the pathophysiology of this disease. This discovery had immediate translational consequences as it provided a druggable target for FOP, leading to clinical development of the investigational Activin A antibody garetosmab [32]. In parallel, it necessitated a reevaluation of the molecular mechanisms that drive HO and other phenotypes of FOP. To begin with, the discovery that HO in FOP requires activation of ACVR1FOP by Activin and that inhibition of Activin A completely blocks HO, dispensed of the notion that hyperresponsiveness to BMP or constitutive activity of ACVR1FOP are driving HO in FOP. Surprisingly, multiple other reviews fail to clarify that HO in FOP does not result from either constitutive activation or hyperresponsiveness of ACVR1FOP to BMPs [33, 34, 77-82] but is rather driven by activation of ACVR1FOP by Activin A. We posit that this perspective is incorrect, as has been demonstrated by our work [29, 37] as well as that of Hino et al [30], has been independently replicated [47, 83-86], and further buttressed by the results of a recent clinical trial [32].
Furthermore, the studies exploring the role of Activin A as well as the utility of anti-ACVR1 antibodies in FOP revealed that activation of ACVR1FOP alone is not adequate to drive the FAPs down an osteogenic lineage – rather, some other factor(s) is required to ‘prime’ the FAP to a state where activation of the Smad1/5/8 pathway is perceived as an osteogenic signal [29, 46, 47, 50, 51]. This finding has not been discussed in any detail elsewhere, and it is important in that identification of the factor that primes the FAPs to respond to Activin A may provide yet another potential therapeutic target in FOP.
Lastly, FOP provided the first physiological evidence that the interaction between Activin A and ACVR1 is not artifactual, and that ACVR1 is indeed a receptor for Activin A, as had been originally described [41-43]. This finding indicated that wild type ACVR1 forms non-signaling complexes (NSCs) with Activin A leading to the discovery that these non-signaling complexes have a tempering effect on the level of HO in FOP [45, 76]. This data also indicates that formation of the heterotetrameric complexes comprised of two type I receptors and two type II receptors along with the naturally dimeric BMP/TGFß family ligands is the sole requirement for signaling; rather an additional mechanism, such as conformational changes must be at play. This possibility has been cursorily discussed in the literature [87] but has not been explored experimentally.
As informative as these discoveries have been for understanding FOP, many new questions emanate from these discoveries. For example, the mechanism by which FAPs are primed to assume an osteogenic fate in response to Smad1/5/8 is unknown. The mechanism by which stoichiometrically identical complexes comprised of ACVR1, type II receptors, and Activin A, and where ACVR1 is either wild type or ACVR1FOP, bring about diametrically opposite outcomes remains to be elucidated. The physiological roles of the NSCs formed by the Activins with wild type ACVR1 remain completely unexplored outside of FOP. These new lines of inquiry are a direct byproduct of the discovery of Activin A's central role in FOP but are likely to provide answers that will inform BMP/TGFß signaling mechanisms in a broader sense.